# Clinical Phenotypes of Atrial Fibrillation and Mortality Risk—A Cluster Analysis from the Nationwide Italian START Registry

**DOI:** 10.3390/jpm12050785

**Published:** 2022-05-12

**Authors:** Daniele Pastori, Emilia Antonucci, Alberto Milanese, Danilo Menichelli, Gualtiero Palareti, Alessio Farcomeni, Pasquale Pignatelli

**Affiliations:** 1Department of Clinical, Internal, Anesthesiological, and Cardiovascular Sciences, Sapienza University of Rome, 00185 Rome, Italy; danilo.menichelli@uniroma1.it (D.M.); pasquale.pignatelli@uniroma1.it (P.P.); 2Arianna Anticoagulazione Foundation, 40138 Bologna, Italy; e.antonucci@fondazionearianna.org (E.A.); gualtiero.palareti@unibo.it (G.P.); 3Department of Public Health and Infectious Diseases, Sapienza University of Rome, 00185 Rome, Italy; alberto.milanese@uniroma1.it; 4Department of Economics and Finance, University of Rome “Tor Vergata”, 00133 Rome, Italy; alessio.farcomeni@uniroma2.it

**Keywords:** atrial fibrillation, all-cause mortality, phenotype, risk factors

## Abstract

Patients with atrial fibrillation (AF) still experience a high mortality rate despite optimal antithrombotic treatment. We aimed to identify clinical phenotypes of patients to stratify mortality risk in AF. Cluster analysis was performed on 5171 AF patients from the nationwide START registry. The risk of all-cause mortality in each cluster was analyzed. We identified four clusters. *Cluster 1* was composed of the youngest patients, with low comorbidities; *Cluster 2* of patients with low cardiovascular risk factors and high prevalence of cancer; *Cluster 3* of men with diabetes and coronary disease and peripheral artery disease; *Cluster 4* included the oldest patients, mainly women, with previous cerebrovascular events. During 9857 person-years of observation, 386 deaths (3.92%/year) occurred. Mortality rates increased across clusters: 0.42%/year (cluster 1, reference group), 2.12%/year (cluster 2, adjusted hazard ratio (aHR) 3.306, 95% confidence interval (CI) 1.204–9.077, *p* = 0.020), 4.41%/year (cluster 3, aHR 6.702, 95%CI 2.433–18.461, *p* < 0.001), and 8.71%/year (cluster 4, aHR 8.927, 95%CI 3.238–24.605, *p* < 0.001). We identified four clusters of AF patients with progressive mortality risk. The use of clinical phenotypes may help identify patients at a higher risk of mortality.

## 1. Introduction

Atrial fibrillation (AF) is a highly prevalent cardiac disease characterized by an increased risk of thromboembolic events and cardiovascular disease, such as myocardial infarction [1]. In addition to ischemic complications, patients with AF experience a high rate of mortality, which is estimated at ≥4%/year [2,3]. Of note, at least one-third of causes of death are related to non-cardiovascular disease [4,5], which are indeed not significantly modified by antithrombotic treatments.

Despite the extensive use of scores for risk stratification in AF, this approach presents several limitations.

Thus, a recent study confirmed that in AF patients the predictive performance of common risk scores against mortality was limited, with c-indexes generally <0.65 [6]. Some other scores are also difficult to calculate, requiring many clinical and laboratory variables and are therefore not easy to be used in daily clinical practice [7,8]. Furthermore, in the need of a simple approach, some important clinical characteristics are often not included in current risk stratification schemes, neglecting some potential important factors that need to be addressed. In many cases, scores are also applied to cohorts with missing variables or with single items not properly calculated as in the case of retrospective registries using codes. Of note, in patients with AF, only one risk score, the BASIC-AF risk score has been proposed to predict mortality in AF patients [9]. However, this score includes imaging and laboratory variables that are not always available for outpatients [9].

Cluster analysis may play a role in overcoming these limitations, especially in the case of overlapping risk factors. Previous studies showed that clustering may allow a better characterization of the disease phenotype in different clinical settings such as heart failure [10,11] and chronic obstructive pulmonary disease [12]. This approach may have an important impact in clinical practices by implementing risk stratification.

The aim of the study was to analyze in patients enrolled in the large cohort of the START (Survey on anTicoagulated pAtients RegisTer) registry, clinical phenotypes of AF by cluster analysis, and the association with mortality risk.

## 2. Materials and Methods

Details of the multicenter nationwide START registry were previously described [13]. Briefly, the START-register is an observational, multicenter, ongoing cohort study that includes patients (≥18 years) who start anticoagulation therapy. The present analysis is limited to patients with non-valvular AF starting oral anticoagulants, either vitamin K antagonists (VKAs) or direct oral anticoagulants (DOACs). Patients treated with low-molecular weight heparin were excluded. Patients with life expectancy <6 months, or non-residents in the participant region, or planning to leave in the next 6 months, were not included in the registry, as well as patients already enrolled in phase II or III clinical studies. Patients enrolled in other observational or phase IV studies were considered eligible for the study.

### 2.1. Ethics

All patients signed an informed written consent at study entry. The registry was approved in October 2011 (ref. 142/2010/0/0ss) by the Ethical Committee of the Institution of the Coordinating Member (University Hospital “S. Orsola-Malpighi”, Bologna, Italy). The study is registered at clinicaltrials.gov identifier: NCT02219984 and is still ongoing/recruitment is still open. The study is conducted according to the declaration of Helsinki.

In particular, the START registry (Survey on anTicoagulated pAtients RegisTer, NCT02219984), is promoted by the Arianna Anticoagulazione Foundation, Bologna. The registry is investigator-driven, non-sponsored, and was approved by the ethics committee of each participating institution (Campus Bio-Medico University of Rome, Italy; Monaldi Hospital and “Luigi Vanvitelli” University of Campania, Italy; “Federico II” University of Naples, Italy; University of Perugia, Italy; University Hospital of Padua, Italy; Sapienza University of Rome, Italy; University of Florence, Italy).

### 2.2. Patient and Public Involvement Statement

Patients or the public were not involved in the design, conduct, reporting, or dissemination plans of our research.

### 2.3. Statistical Analysis

Data are expressed as mean and standard deviation or the median and interquartile range (IQR) depending on the variable distribution. Group comparisons were performed by the unpaired Student’s *t*-test. Proportions and categorical variables were tested by the χ^2^ test.

In order to identify subgroups of patients with the most similar baseline characteristics we selected a pool of variables and proceeded with cluster analysis. We decided to use the following clinical variables: age, sex, diabetes, previous cerebrovascular events (defined as ischemic stroke or transient ischemic attack), previous cardiovascular events, heart failure, peripheral artery disease (PAD), use of non-vitamin K oral anticoagulants, cancer, pulmonary disease, smoking habit, previous major bleeding. The following clinical variables were instead not used for the cluster analysis: persistent/permanent AF, body mass index (BMI), hypertension. We excluded these variables as their use would have led to a large number of groups, which would not have been useful for clinical purposes. Concomitant drugs were not used for clustering to avoid bias by indication. Furthermore, composite variables, such as CHA_2_DS_2-_VASc and HAS-BLED scores were not used since we used their components. For cluster analysis we proceeded using a model based procedure where continuous variables in each cluster were assumed to follow the multivariate Gaussian distribution, and categorical variables to follow a multinomial distribution, as in Hennig and Liao [14]. Model-based clustering allows us to use a formal criterion for selecting the optimal number of groups. In this work, we selected the optimal number of clusters by comparing minimizing the Bayesian information criterion.

The incidence of all-cause mortality by each cluster was estimated using a Kaplan–Meier product-limit estimator. Survival curves were formally compared using the log-rank test. Univariable and multivariable Cox proportional hazard regression analyses were used to calculate the (adjusted) relative hazard ratios (HRs) of death. In the multivariable model, we adjusted for variables not used to define the clusters, not using composite variables (such as risk scores) to avoid overadjustment.

All tests were two-tailed, a *p*-value < 0.05 was considered statistically significant. Analyses were performed using computer software IBM^®^ SPSS^®^ Statistics version 25 (IBM, Armonk, NY, USA). and R version 3.6.2 (RStudio, Boston, MA, USA).

## 3. Results

### 3.1. Description of Clusters

Table 1 shows clinical and biochemical characteristics of patients according to each cluster.

**Cluster 1 (*n* = 512). Youngest and low comorbidities.** This cluster included patients with the lowest mean age (55.6 ± 7.9 years) and with a low prevalence of women (23.6%), and with the overall lowest burden of cardiovascular comorbidities with only 14.6% of patients with a history of cerebrovascular events (second lowest prevalence). These patients were less likely to be treated with DOACs (only 10%). In this cluster, there was the highest proportion of obese patients (30.1%) and the highest use of anti-arrhythmic drugs (32.8%), probably related to the low proportion of patients with persistent/permanent AF (51.4%).

**Cluster 2 (*n* = 2201). Low cardiovascular risk and high cancer.** This was the largest cluster including patients with a relatively high mean age (75.0 ± 6.0 years) and 54% of patients were women (second highest group). This group was characterized by the lowest prevalence of cardiovascular risk factors and the highest proportion of patients with cancer (18.2%). Regarding anticoagulant treatment, the use of DOACs was present in 27% of patients. Among medications, aspirin was prescribed in 5.2% of patients despite a very low prevalence of cardiovascular disease at baseline (1.5%). The prevalence of anemia was significantly higher than cluster 1 (19.3% versus 11.3%, respectively).

**Cluster 3 (*n* = 1268). High cardiovascular risk and more men.** This cluster displayed the lowest prevalence of women (8.1%) while the mean age was similar to cluster 2. Cardiovascular risk factors were highly prevalent, being the cluster with the highest proportion of diabetes (35.0%), previous cardiovascular disease (53.5%), PAD (16.1%), and chronic pulmonary disease (27.8%). Moreover, previous cerebrovascular events (17.2%) and heart failure (29.2%) were common, with the second highest prevalence among clusters. This cluster disclosed the highest use of aspirin in 21.1%. Among variables not used for clustering, this group was the first for highest for the use of proton pump inhibitors (58.7%) and statins (54.7%) and for the prevalence of thrombocytopenia (14.7%), and the second highest for the prevalence of anemia and hypertension.

**Cluster 4 (*n* = 1190). Oldest, more women, and cerebrovascular disease.** This cluster was composed mainly by elderly patients (mean age 83.7 ± 4.2) with the highest number of women (78.2%) and persistent/permanent AF (70.8%). The prevalence of previous cerebrovascular events in this group was the highest among clusters with 23.9% of patients, as well as heart failure (31.1%); the second highest proportion of chronic pulmonary disease (21.0%) and previous cardiovascular disease (18.1%) was found. This was the group with the highest use of DOACs (33.8%). Regarding other comorbidities, this group had the highest prevalence of chronic kidney disease (78.8%), anemia (34.6%), and hypertension (88.2%). Concerning medications, this cluster had the highest use of DOACs (33.8%), digoxin (15.8%), and the second highest use of statins (29.5%) and proton pump inhibitors (53.1%).

### 3.2. Clusters and Mortality Risk

During a mean follow-up of 22.9 ± 16.7 months yielding 9857 person-years of observation, 386 deaths (3.92%/year) were registered. Incidence rate of mortality was 0.42%/year (95%CI 0.11–1.10) in cluster 1 (reference group), 2.12%/year (95%CI 1.71–2.59) in cluster 2 (HR 5.068, 95%CI 1.863–13.784, *p* = 0.001 versus cluster 1), 4.41%/year (95%CI 3.60–5.35) in cluster 3 (HR 10.513, 95%CI 3.872–28.544, *p* < 0.001 versus cluster 1), 8.71%/year (95%CI 7.50–10.1) in cluster 4 (HR 20.708, 95%CI 7.690–55.761, *p* < 0.001 versus cluster 1) (Figure 1).

Kaplan–Meier curves (Figure 2) showed a significant difference in the incidence of mortality across clusters, which increased from cluster 1 to 4 (log-rank test *p* < 0.001).

At the multivariable Cox proportional regression analysis, clusters remained associated with mortality after adjustment for confounding factors and medications (Table 2).

## 4. Discussion

From the large dataset of the Italian START registry, we identified four groups of AF patients with specific characteristics and graded progressive risk of all-cause mortality.

The four clusters showed specific clinical characteristics. Cluster 1 was the group with the lowest incidence of mortality and was composed of the youngest patients, with obesity and low comorbidities. This group was coincidentally characterized by a relatively lower proportion of paroxysmal AF, compared to the other clusters. The lower risk of mortality in patients with paroxysmal AF was reported in the post hoc analysis of the ENGAGE AF-TIMI 48 Trial (Effective Anticoagulation with Factor Xa Next Generation in Atrial Fibrillation-Thrombolysis in Myocardial Infarction 48), which showed a lower risk of mortality in paroxysmal versus permanent AF (1.49%/year and 1.95%/year, respectively) [15]. This cluster was also characterized by a higher proportion of obese patients. Regarding obesity, the real meaning of this association is hard to explain, as BMI does not take into account fat composition and visceral adiposity. A recent analysis showed a U-shaped association between body weight and mortality in AF [16].

Cluster 2 included patients with low cardiovascular risk factors and a high proportion of cancer. Patients in this group disclosed a five-fold increased risk of mortality compared to patients without cancer. This finding is in line with a previous finding showing that cancer is an important risk factor for mortality in the AF population [5,17], requiring specific management of anticoagulation according to cancer-specific treatments [18].

Cluster 3 was composed of mainly men with diabetes and coronary and PAD, a high proportion of thrombocytopenia, and a high use of aspirin, proton pump inhibitors, and statins. This cluster clearly defines patients with vascular disease. Coronary disease and PAD are frequently associated in patients with AF and increase the risk of cardiovascular events [19]. Thus, it is not surprising that in this cluster there was a high use of statins, which are recommended to prevent cardiovascular events in patients with PAD [20] and have been shown to improve outcomes in the AF population [21].

Cluster 4 included the oldest patients, mainly women, with previous cerebrovascular events, persistent/permanent AF, heart failure, kidney disease and anemia. The lowest prevalence of obesity in this cluster may reflect the association of sarcopenia with advancing age. The high proportion of patients with heart failure in this cluster (>30%) confirms the pivotal role of this comorbidity as the leading cause of death in patients with AF, even more important than ischemic stroke [22,23].

Our analysis has significant differences when compared to two previous studies that analyzed the clinical phenotype of AF patients [24,25]. The first study developed clusters from a Japanese cohort with very different clinical characteristics than our cohort, such as low prevalence of hypertension, which is actually one of the leading causes of mortality worldwide [24]. Furthermore, about 11% of patients with an indication to oral anticoagulation, i.e., a CHA_2_DS_2-_VASc score equal to or above 2, were not taking anticoagulants [26]. Another study included 9749 patients with AF in the US [25], and identified four clusters which, however, were not easy to use; thus, cluster 3 shared similar characteristics of cluster 4 regarding the proportion of hypertension, respiratory and chronic kidney disease, along with a similar age [25]. Furthermore, the proportion of diabetes was very similar between clusters 3 and 2 [25]. This overlap of risk factors was also evident in the external cohort from the ORBIT-AF [25]. All these factors make the allocation of patients in a specific cluster difficult. Finally, cancer was not considered for cluster formation in either of the two studies; this is an important point as we believe that it may define a specific subgroup of AF patients with peculiar characteristics.

The identification of clinical clusters of AF patients at different mortality risks may be complementary to the integrated approach proposed by recent guidelines for the management of AF patients [26,27]. In this view, the application of the ABC pathway may differ in the four clusters. For instance, patients in cluster 1 may benefit from an early rhythm and symptoms control, whilst patients in clusters 3 and 4 from a tight control of cardiometabolic diseases, such as diabetes, hypertension, and dyslipidemia. This tailored approach may lead to a reduction of mortality in this population of high-risk patients.

Our study has some strengths and limitations to acknowledge. Cluster analysis is an unbiased approach to identify patients at a higher risk of death. Thus, apart from oral anticoagulation, which is a common indication for all patients, we did not use concomitant drugs to define clusters, to avoid any bias by indication. Moreover, we also considered risk factors not included in the current risk stratification scores, providing information on additional comorbidities that need to be managed in AF patients. The large sample size of the study cohort, which recruited patients from any region of our country, is another strength of our work, which makes our study representative of our general AF population and adequate to perform the cluster analysis. Despite the advantage provided by a cluster-based approach to identify subgroups of patients within a specific disease, the problem with the cluster analysis is the generalization of results to other populations [28].

As a limitation, and an open field for future research, we could not investigate the association of clusters with specific causes of death. We only included patients of Caucasian ethnicity; thus, clinical phenotypes may be different in other populations.

In conclusion, we identified specific phenotypes of AF patients showing a different association with mortality. A correct global risk stratification strategy should include clinical phenotypes of patients beyond risk scores application.

## Figures and Tables

**Figure 1 jpm-12-00785-f001:**
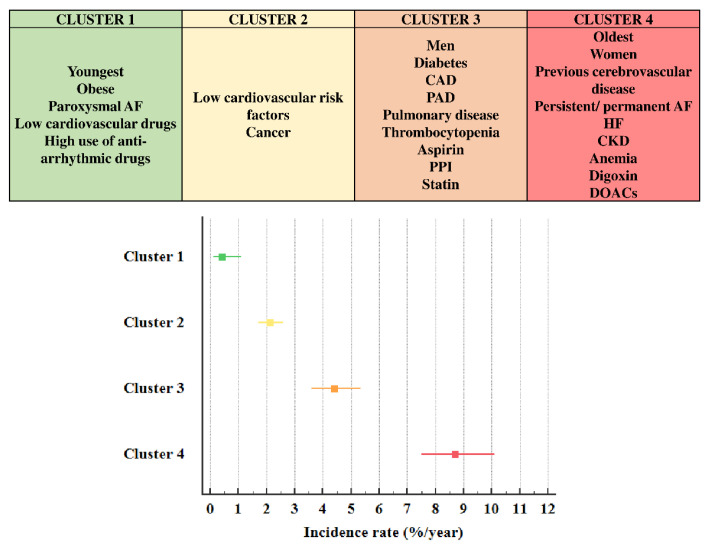
Description of clusters characteristics and incidence rates of mortality. Abbreviations: AF: atrial fibrillation; CAD: coronary artery disease; PAD: peripheral artery disease; PPI: proton pump inhibitors; HF: heart failure; CKD: chronic kidney disease; DOACs: direct oral anticoagulants.

**Figure 2 jpm-12-00785-f002:**
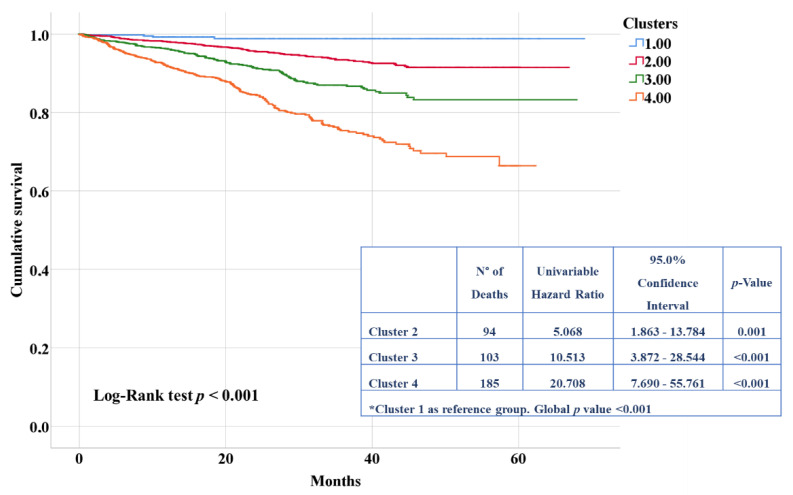
Kaplan–Meier curves for risk of mortality according to different clusters.

**Table 1 jpm-12-00785-t001:** Clinical and biochemical characteristics of patients according to each cluster.

Cluster Denomination	Whole Cohort	Cluster 1	Cluster 2	Cluster 3	Cluster 4	*p*-Value (among Groups)
		Youngest and Low Comorbidities	Low Cardiovascular Risk and High Cancer	High Cardiovascular Risk and More Men	Oldest, More Women and Cerebrovascular Disease	
Cluster size *n*	5171	512	2201	1268	1190	
	Variables used to define clusters					
Age (years)	75.0 ± 9.6	55.6 ± 7.9	75.0 ± 6.0	74.6 ± 7.0	83.7 ± 4.2	<0.001
Women (%)	45.3	23.6	54.0	8.1	78.2	<0.001
Diabetes (%)	20.2	10.7	16.5	35.0	15.1	<0.001
Previous cerebrovascular events (%)	16.5	14.6	12.5	17.2	23.9	<0.001
Previous cardiovascular disease (%)	18.6	6.8	1.5	53.5	18.1	<0.001
Heart failure (%)	15.5	7.0	1.1	29.2	31.1	<0.001
Peripheral Artery Disease (%)	6.4	0.8	0.6	16.1	9.1	<0.001
Cancer (%)	13.6	2.9	18.2	15.1	8.1	<0.001
Pulmonary disease (%)	12.6	3.1	1.5	27.8	21.0	<0.001
Smoking (%)	13.2	21.9	2.7	39.4	1.1	<0.001
Previous major bleeding (%)	3.5	1.4	1.9	4.5	6.1	<0.001
DOACs (vs. VKAs) (%)	25.8	10.0	27.0	22.7	33.8	<0.001
	Variables not used for cluster analysis					
Persistent/permanent AF (%)	63.3	51.4	60.7	65.7	70.8	<0.001
BMI (kg/m^2^)	26.9 ± 4.7	28.1 ± 5.5	26.7 ± 4.5	27.6 ± 4.6	25.8 ± 4.6	<0.001
Obesity (BMI ≥ 30 kg/m^2^)	21.1	30.1	19.9	24.1	16.6	<0.001
Creatinine Clearance (mL/min)	66.8 ± 28.3	103.8 ± 33.6	67.6 ± 22.8	68.6 ± 26.8	47.6 ± 17.4	<0.001
Chronic kidney disease (Creatinine clearance <60 mL/min) (%)	45.1	5.1	39.5	39.4	78.8	<0.001
Hemoglobin (g/dl)	13.5 ± 1.8	14.5 ± 1.6	13.6 ± 1.6	13.6 ± 1.8	12.7 ± 1.6	<0.001
Anemia (<12 g/dL for women and <13 g/dL for men) (%)	24.7	11.3	19.3	30.0	34.6	<0.001
Platelet count (×10^9^/L)	222.2 ± 68.9	223.3 ± 62.0	223.0 ± 69.6	213.8 ± 70.7	229.2 ± 67.7	<0.001
Thrombocytopenia (<150 × 10^9^/L, %)	10.7	9.0	10.3	14.7	7.9	<0.001
Hypertension (%)	80.6	59.6	78.1	86.3	88.2	<0.001
CHA_2_DS_2_ VASc score	3.6 ± 1.5	1.5 ± 1.1	3.3 ± 1.2	3.9 ± 1.4	4.7 ± 1.2	<0.001
HAS-BLED score	1.3 ± 0.7	0.4 ± 0.6	1.2 ± 0.6	1.5 ± 0.8	1.5 ± 0.6	<0.001
Aspirin (%)	9.7	6.3	5.2	21.1	7.5	<0.001
Statins (%)	33.7	21.3	26.8	54.7	29.5	<0.001
Anti-arrhythmic drugs (%)	25.2	32.8	26.2	25.1	20.3	<0.001
Digoxin (%)	9.2	6.1	7.2	8.0	15.8	<0.001
Proton pump inhibitors (%)	45.9	32.6	37.8	58.7	53.1	<0.001

BMI: body mass index; DOAC: direct oral anticoagulant; VKA: vitamin K antagonist; AF: atrial fibrillation; BMI, body mass index.

**Table 2 jpm-12-00785-t002:** Multivariable Cox proportional regression analysis of factors associated with mortality.

Variables	Hazard Ratio	95% Confidence Interval		*p*-Value
**Cluster 2 (vs. 1) ***	**3.306**	**1.204**	**9.077**	**0.020**
**Cluster 3 (vs. 1) ***	**6.702**	**2.433**	**18.461**	**<0.001**
**Cluster 4 (vs. 1) ***	**8.927**	**3.238**	**24.605**	**<0.001**
Persistent/permanent AF	1.231	0.975	1.553	0.081
**Statin**	**0.655**	**0.519**	**0.828**	**<0.001**
Digoxin	0.963	0.692	1.339	0.822
**Proton pump inhibitors**	**1.367**	**1.108**	**1.686**	**0.004**
Hypertension	1.009	0.747	1.363	0.953
Obesity	1.217	0.930	1.592	0.152
**Anemia**	**1.618**	**1.313**	**1.993**	**<0.001**
**Thrombocytopenia**	**1.418**	**1.060**	**1.898**	**0.019**
**Chronic kidney disease**	**2.347**	**1.821**	**3.024**	**<0.001**
**Anti-arrhythmic drugs**	**0.713**	**0.552**	**0.922**	**0.010**
Aspirin	0.880	0.620	1.248	0.472

* Global *p*-value *p* < 0.001. Abbreviation: AF: atrial fibrillation. Statistically significant values are marked with bold

## Data Availability

The data underlying this article will be shared upon reasonable request to the corresponding author.

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
