# Peer review of "Clinical Phenotypes of Atrial Fibrillation and Mortality Risk—A Cluster Analysis from the Nationwide Italian START Registry"

_jpm, 2022, doi:10.3390/jpm12050785_

Round 1

Reviewer 1 Report

In this interesting paper, Pastori et al. report on research focusing around cluster analysis of the START registry database with an aim to yield high-mortality risk phenotypes in AF. My remarks can be found below.

Abstract

Introduction

The introduction is well-written and provides all necessary information for non-specialists to understand both the significance of risk stratification in AF as well as the promise of cluster analysis in medicine.

However, I did notice a trend for overt self-citation by the authors. In several parts of this manuscript, citing previous work of their own is completely understandable, as it is actually significant previous research on the topic at hand. That said, there are two instances that where the authors stray into the unethical. The first happens in the Introduction section – reference number [7] is completely unrelated to the sentence in which it is cited – “Despite the extensive use of scores for risk stratification in AF”. This could be an honest mistake – however, keep in mind the fact that over 30% of refs in this paper are self-citations by the authors. Ref [31] is also questionable – here it is pertinent to the text, but it is obvious that the authors chose to mention the possibility of left atrial size having a bearing on AF mortality in order to cite their own work. In my opinion, the authors have significant research that is pertinent to this work that should be cited – however, they need to critically re-examine the entire text and remove self-citations of questionable relation to the manuscript.

In p.1 line 37 consider rephrasing “Thus, most scores do not achieve an enough predictive accuracy…” to “Thus, most scores do not achieve acceptable predictive accuracy,…”.

In p.1 line 37-39 the authors state that “Despite the extensive use of scores for risk stratification in AF[7], this approach presents several limitations. Thus, most scores do not achieve an enough predictive accuracy, and prediction models do not usually perform well when they are applied in populations with very different characteristics than the original derivation cohorts”. This paragraph feels out of place, since it mentions limitations and poor predictive accuracy that are only really introduced in the next paragraph (“Thus, a recent study confirmed that in AF patients the predictive performance of common risk scores against mortality was limited, with c-indexes generally <0.65[8]. Some other scores are also difficult to calculate requiring many clinical and laboratory variables and are therefore not easy to be used in daily clinical practice”). Consider reworking this part of the introduction.

In p.2 lines 48-49 the authors claim that no risk score has been developed to predict all-cause death in AF. This is untrue – I kindly recommend the authors familiarize themselves with the BASIC-AF risk score and discuss it (10.1016/j.hjc.2021.01.007).

Methods

The methodology of this cluster analysis checks out, and the Methods section of the manuscript is well-written. However, there is one limitation to the authors’ methodology – the lack of a validation set. Since the authors didn’t split the registry dataset to training and validation set, it is unsure to what extent the findings of the present study are generalizable to external datasets (see 10.1080/09638237.2018.1437615). The authors should mention this in the limitations section.

Results

The results section is also ok in general. Please introduce a column in Table 1 that shows the clinical characteristics of the whole sample – this way the reader can intuitively understand how much each cluster deviates from the norm. On that topic, I find that the authors’ decision to exclude BMI from the cluster analysis is regrettable, since such a stark difference in BMI is noted in the final clusters. Is it possible that cluster analysis could be even better at producing high-risk cohorts if it was allowed BMI information? The authors should discuss this possibility.

Furthermore, I think that the denominations given by authors to each cluster greatly aid reader comprehension, but may also lead readers to false conclusions. By using these denominations, the reader may be led to believe that they are absolute e.g., that all cancer patients were in Cluster 2 or that Cluster 2 only contained cancer patients. That is untrue – in fact, only 41% of cancer patients are in Cluster 2, such that it may be the most common cluster for cancer patients, but the majority lie outside of it. For comparison please refer to similar cluster analyses such as (https://mdpi-res.com/d_attachment/healthcare/healthcare-10-00268/article_deploy/healthcare-10-00268-v2.pdf). In this example, the authors might refer to a cluster as “regular alcohol drinkers” – but that is because that cluster contains 100% of regular drinkers! The authors should reconsider their approach to cluster naming – either remove it or make it absolutely clear that these are names purely created for ease of comprehension and do not match up with real cluster make-up.

Discussion

Discussion of each cluster should be reflective of their actual make-up – e.g., the authors state that Cluster 1 contained mostly patients with paroxysmal AF, but in reality 51.4% of patients had persistent AF!

General remarks

To my understanding, this is an interesting cluster analysis with potential clinical implications. However, there are two drawbacks – firstly, the lack of a validation set precludes any discussion of generalizability to other datasets. Secondly, the fact that the resultant clusters were not clearly delineated makes it difficult to draw external conclusion. These are limitations that should be made clear to readers.

Author Response

Introduction

The introduction is well-written and provides all necessary information for non-specialists to understand both the significance of risk stratification in AF as well as the promise of cluster analysis in medicine.

Response: Thank you for your comment

__________________________________________________________________________________

However, I did notice a trend for overt self-citation by the authors. In several parts of this manuscript, citing previous work of their own is completely understandable, as it is actually significant previous research on the topic at hand. That said, there are two instances that where the authors stray into the unethical. The first happens in the Introduction section – reference number [7] is completely unrelated to the sentence in which it is cited – “Despite the extensive use of scores for risk stratification in AF”. This could be an honest mistake – however, keep in mind the fact that over 30% of refs in this paper are self-citations by the authors. Ref [31] is also questionable – here it is pertinent to the text, but it is obvious that the authors chose to mention the possibility of left atrial size having a bearing on AF mortality in order to cite their own work. In my opinion, the authors have significant research that is pertinent to this work that should be cited – however, they need to critically re-examine the entire text and remove self-citations of questionable relation to the manuscript.

Response: We fully agree with your comment. We are sorry for that. The manuscript has been revised by different colleagues before submission and this resulted in an unnecessary over self-citation rate. Ref 7 was a mere mistake. I have revised the manuscript and removed some references (1-7-19-30-31)

__________________________________________________________________________________

In p.1 line 37 consider rephrasing “Thus, most scores do not achieve an enough predictive accuracy…” to “Thus, most scores do not achieve acceptable predictive accuracy,…”.

AND

In p.1 line 37-39 the authors state that “Despite the extensive use of scores for risk stratification in AF[7], this approach presents several limitations. Thus, most scores do not achieve an enough predictive accuracy, and prediction models do not usually perform well when they are applied in populations with very different characteristics than the original derivation cohorts”. This paragraph feels out of place, since it mentions limitations and poor predictive accuracy that are only really introduced in the next paragraph (“Thus, a recent study confirmed that in AF patients the predictive performance of common risk scores against mortality was limited, with c-indexes generally <0.65[8]. Some other scores are also difficult to calculate requiring many clinical and laboratory variables and are therefore not easy to be used in daily clinical practice”). Consider reworking this part of the introduction.

Response: We removed the sentence starting with “Thus, most scores…”.

__________________________________________________________________________________

In p.2 lines 48-49 the authors claim that no risk score has been developed to predict all-cause death in AF. This is untrue – I kindly recommend the authors familiarize themselves with the BASIC-AF risk score and discuss it (10.1016/j.hjc.2021.01.007).

Response: Thank you for this suggestion. We have mentioned this work in the introduction.

__________________________________________________________________________________

Methods

The methodology of this cluster analysis checks out, and the Methods section of the manuscript is well-written. However, there is one limitation to the authors’ methodology – the lack of a validation set. Since the authors didn’t split the registry dataset to training and validation set, it is unsure to what extent the findings of the present study are generalizable to external datasets (see 10.1080/09638237.2018.1437615). The authors should mention this in the limitations section.

Response: This is a very interesting and appropriate manuscript. Thank you. We have discussed this point in the limitations section.

__________________________________________________________________________________

Results

The results section is also ok in general. Please introduce a column in Table 1 that shows the clinical characteristics of the whole sample – this way the reader can intuitively understand how much each cluster deviates from the norm. On that topic, I find that the authors’ decision to exclude BMI from the cluster analysis is regrettable, since such a stark difference in BMI is noted in the final clusters. Is it possible that cluster analysis could be even better at producing high-risk cohorts if it was allowed BMI information? The authors should discuss this possibility.

Response: We have added data on the whole cohort as suggested. This makes the differences even more evident, thank you. Please see new table 1.

The use of BMI was not considered for cluster formation as it did not lead to an improvement in the discrimination of clusters against mortality in this cohort.

__________________________________________________________________________________

Furthermore, I think that the denominations given by authors to each cluster greatly aid reader comprehension, but may also lead readers to false conclusions. By using these denominations, the reader may be led to believe that they are absolute e.g., that all cancer patients were in Cluster 2 or that Cluster 2 only contained cancer patients. That is untrue – in fact, only 41% of cancer patients are in Cluster 2, such that it may be the most common cluster for cancer patients, but the majority lie outside of it. For comparison please refer to similar cluster analyses such as (https://mdpi-res.com/d_attachment/healthcare/healthcare-10-00268/article_deploy/healthcare-10-00268-v2.pdf). In this example, the authors might refer to a cluster as “regular alcohol drinkers” – but that is because that cluster contains 100% of regular drinkers! The authors should reconsider their approach to cluster naming – either remove it or make it absolutely clear that these are names purely created for ease of comprehension and do not match up with real cluster make-up.

Response: You are completely right. This was made in an attempt to make the identification of clusters easier for the readers. I have amended the label of each cluster into a more descriptive form to make the description of clusters clearer as follows:

Cluster 1: Youngest and low comorbidities       

Cluster 2: Low cardiovascular risk and high cancer        

Cluster 3: High cardiovascular risk and more men

Cluster 4: Oldest, more women and cerebrovascular disease

__________________________________________________________________________________

Discussion

Discussion of each cluster should be reflective of their actual make-up – e.g., the authors state that Cluster 1 contained mostly patients with paroxysmal AF, but in reality 51.4% of patients had persistent AF!

Response: Amended as suggested.

__________________________________________________________________________________

General remarks

To my understanding, this is an interesting cluster analysis with potential clinical implications. However, there are two drawbacks – firstly, the lack of a validation set precludes any discussion of generalizability to other datasets. Secondly, the fact that the resultant clusters were not clearly delineated makes it difficult to draw external conclusion. These are limitations that should be made clear to readers.

Response: Thank you for your thorough revision of our work and the very useful comments. The manuscript significantly improved after revision.

Reviewer 2 Report

The article “Clinical Phenotypes of Atrial Fibrillation and Mortality Risk. A 2 Cluster Analysis from the Nationwide Italian START Registry” fills a gap in knowledge regarding specific risk of mortality in atrial fibrillation patients.

The article is well written and documented and sets the stage for further research. The methodology is innovative and the conclusions are pertinent. The tables and figures adequately illustrate the results. However the number of self-citations is somehow excessive, one or two of them being over the counter.

Author Response

Reviewer 2

The article “Clinical Phenotypes of Atrial Fibrillation and Mortality Risk. A 2 Cluster Analysis from the Nationwide Italian START Registry” fills a gap in knowledge regarding specific risk of mortality in atrial fibrillation patients. The article is well written and documented and sets the stage for further research. The methodology is innovative and the conclusions are pertinent. The tables and figures adequately illustrate the results.

Response: Thank you for your comment

__________________________________________________________________________________

However the number of self-citations is somehow excessive, one or two of them being over the counter.

Response: As mentioned in the responses to Reviewer #1, we fully agree with your comment and we are sorry for that. The manuscript has been revised by different colleagues before submission and this resulted in an unnecessary over self-citation rate. I have revised the manuscript and removed some references (1-7-19-30-31)

Round 2

Reviewer 1 Report

The authors correctly addressed previous comments - the article is now worthy of publication.